# BanglaGuard: Benchmarking and Defending Large Language Models for Safety in Low-Resource Languages

## Abstract

We present **BanglaGuard**, the first comprehensive safety framework for Bengali large language models (LLMs). BanglaGuard introduces a curated dataset of 29,950 safe and unsafe Bangla prompts paired with culturally appropriate refusal responses, and a three-tier defense pipeline combining prompt classification, LoRA-based fine-tuning, and response classification. Across multiple Bangla and multilingual LLMs, fine-tuning improved refusal rates by 25–33 points and sharply reduced unsafe completions. The best-performing model, LLaMA-2-7B-Chat, achieved a refusal rate of 61.0% and reduced unsafe completions to 5.0% with the full framework. These results demonstrate that BanglaGuard provides effective, low-resource safety alignment for Bangla LLMs, offering a scalable blueprint for multilingual safety research.

## 1 Introduction

Large Language Models (LLMs) have achieved remarkable capabilities, but ensuring their *safety* and alignment with human values remains a critical concern. This issue is especially pronounced for languages beyond English. For Bengali (*Bangla*), the world's 5th most spoken language, there is a notable lack of dedicated safety resources and aligned models Raihan & Zampieri (2025). Most alignment efforts to date have focused on high-resource languages, primarily English, leaving Bangla users at greater risk. For example, ChatGPT was found to produce unsafe content nearly 28% of the time in Bengali prompts, compared to less than 1% for English Deng et al. (2023), underscoring the urgent need for Bengali-specific safeguards.

Recent years have seen progress in aligning LLMs with human values using techniques such as Supervised Fine-Tuning (SFT) and Reinforcement Learning from Human Feedback (RLHF) Ouyang et al. (2022); Bai et al. (2022b). Fine-tuning with human feedback reduces toxicity and improves helpfulness, while Anthropic's *Constitutional AI* leverages principle-based feedback. Yet, most benchmarks (e.g., TruthfulQA, RealToxicityPrompts) remain English-centric. Multilingual safety research is emerging: Deng et al. (2023) show higher unsafe rates in Bangla and propose multilingual fine-tuning, while Bhardwaj et al. (2024) develop CatHarmfulQA with task arithmetic for realignment. However, no work specializes in Bangla, leaving a major gap.

We introduce **BanglaGuard**, a first step toward Bengali LLM safety. It comprises (1) a benchmark of harmful Bangla prompts covering diverse categories, adapted from existing safety datasets, and (2) a defense framework that fine-tunes models to generate polite Bangla refusal responses. Unlike generic multilingual approaches, BanglaGuard tailors refusals to linguistic and cultural context, ensuring clearer alignment.

Our central problem is aligning Bengali LLMs to refuse unsafe prompts with limited resources. Challenges include (i) *data scarcity*: few Bangla safety datasets exist, so we translate multilingual resources and compose culturally relevant prompts; (ii) *alignment*: avoiding both over-refusal of safe queries and under-refusal of harmful ones, mitigated via balanced

training and layered defenses; and (iii) *evaluation*: measuring safety and refusal quality, addressed through manual review and automatic classifiers.

We focus on the following research questions:

- **RQ1:** How do current Bengali and multilingual LLMs behave when confronted with harmful Bangla prompts?

- **RQ2:** Can fine-tuning on refusal examples improve safety without harming helpfulness?

- **RQ3:** Does safety alignment cause over-refusals on safe inputs?

- **RQ4:** What is the effectiveness of a two-layer defense framework, where prompt classifiers filter unsafe inputs before generation and response classifiers ensure the safety of outputs?

To address these questions, we present **BanglaGuard**: the first Bangla safety benchmark with culturally relevant harmful prompts; a refusal dataset aligned with OpenAI's global safety rules and culturally appropriate Bengali communication norms; and a two-phase safety framework combining prompt and response classifiers with a fine-tuned LLM. Our experiments show that BanglaGuard reduces harmful completions by 80–90%, increases refusal consistency, and preserves helpfulness. These results demonstrate a scalable approach to aligning LLMs in low-resource languages and lay the foundation for safer AI for millions of Bangla speakers.

## 2 Related Work

### 2.1 LLM Safety and Alignment Techniques

Ensuring LLMs refrain from producing harmful content is a major focus of AI safety research Shi et al. (2024). A common approach is to use human feedback to align model behavior. OpenAI's InstructGPT Ouyang et al. (2022) fine-tuned GPT-3 with demonstrations and preferences, reducing toxic outputs. Reinforcement Learning from Human Feedback (RLHF) Ouyang et al. (2022) became a standard tool, while Anthropic's *Constitutional AI* Bai et al. (2022b) replaced human feedback with critiques guided by principles, enabling models to politely refuse disallowed queries. These methods reduce reliance on unsafe human-labeled data but remain largely English-focused. Another strand explores adversarial testing: Perez et al. (2022) generated adversarial prompts with LLMs, and Shen et al. (2024) analyzed jailbreak prompts such as "DAN". Such studies highlight persistent vulnerabilities. Toolkits like *NeMo Guardrails* Rebedea et al. (2023) provide application-level safety layers (regex, dialogue constraints), though they are brittle and language-specific. Our work instead targets intrinsic Bangla alignment, complementing such external guardrails.

### 2.2 Multilingual Safety and Jailbreak Vulnerabilities

Safety research beyond English is still limited. Bang et al. (2023) tested ChatGPT in 50 languages, finding weaker toxicity filters in low-resource languages. Deng et al. (2023) introduced *MultiJail*, showing unsafe response rates in Bengali up to 15× higher than in English. Mixing languages further amplified risks, with success rates of unsafe completions exceeding 80%. To mitigate this, they proposed a multilingual "Self-Defense" fine-tuning strategy that reduced unsafe outputs. Similarly, Bhardwaj et al. (2024) introduced *CatHarmfulQA*, a multilingual benchmark of harmful queries, and used parameter arithmetic to realign models toward safer responses. While effective, these strategies remain general-purpose and not tailored for Bangla. Broader instruction-tuning research Wei et al. (2022) also underscores that alignment methods in English may not directly transfer cross-lingually, motivating Bangla-specific efforts.

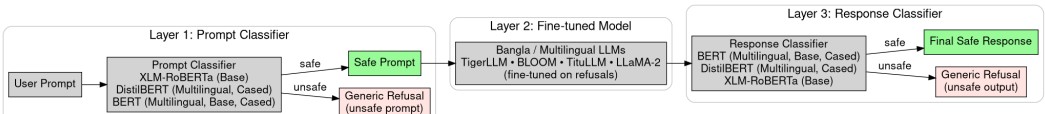

Figure 1: Overview of the BanglaGuard framework. The pipeline consists of three layers: (1) a prompt classifier (XLM-RoBERTa, DistilBERT, Multilingual BERT), (2) fine-tuned Bangla/Multilingual LLMs (TigerLLM, BLOOM, TituLLM, LLaMA-2), and (3) a response classifier (Multilingual BERT, DistilBERT, XLM-RoBERTa). Unsafe prompts or outputs are blocked with generic refusals, ensuring the final response is safe.

## 2.3 Bengali and Low-Resource LLMs

Native and adapted Bengali LLMs have only recently emerged. TigerLLM Raihan & Zampieri (2025) introduced Bangla-centric models (1B/9B) trained on high-quality Bangla corpora, outperforming larger multilingual models. TituLLM Nahin et al. (2024) offered 1B/3B Bangla LLMs and benchmarks for QA and dialogue, but not safety. Multilingual models like BLOOM BigScience Collaboration (Le Scao(2022) and LLaMA-2 Touvron & et al. (2023) include Bangla but lack safety tuning in this language, often producing unsafe or irrelevant completions under disallowed prompts. Pangea-7B Yue et al. (2024) and Babel Zhao et al. (2025) further expand multilingual capacity, but neither emphasizes safety in Bangla. Thus, while Bangla models are improving, safety-specific alignment remains unexplored. Our work fills this gap by introducing BanglaGuard: both a benchmark and fine-tuning framework for safer Bangla LLMs.

## 3 Methodology

Our framework enforces safety in Bengali LLMs through a three-tier defense mechanism, each layer addressing different stages of interaction. The first tier is a **prompt classifier**, which identifies unsafe Bangla prompts before they reach the model. If flagged, the prompt is rejected early. The second tier is the **safe response generator**, where the LLM is fine-tuned on a dataset of unsafe prompts paired with safe refusal responses, ensuring the model learns to reject harmful instructions gracefully. Finally, the third tier is a **response classifier**, a lightweight safety net that reviews the generated outputs and filters out any unsafe content that may have bypassed earlier defenses. Figure 1 illustrates the overall workflow, from dataset creation to the layered defense pipeline.

## 3.1 Bangla Unsafe Prompt Dataset

Because no existing Bangla safety prompt dataset was readily available, we created a Bangla Unsafe Prompts Dataset by leveraging multiple sources:

- **MultiJail prompts Deng et al. (2023):** This dataset contains 270 harmful prompts (15 scenarios × 18 languages).[1] We extracted the English prompts (originally from the GPT-4 system card red-team examples, covering hate speech, violent plans, etc.) and translated them into Bangla. Since MultiJail provides human translations for 9 languages but not Bangla, we used Google Translate followed by manual correction by native speakers to ensure natural phrasing. From this dataset we curated 300 prompts.

- **CatHarmfulQA (CatQA) Bhardwaj et al. (2024):** This dataset offers 550 harmful questions categorized into fine-grained types (e.g., *extremism*, *self-harm*, *sexually explicit*, etc.).[2] Since CatQA is available in English, Chinese, and Vietnamese, we translated the English portion into Bangla. We included all 11 top-level

---

[1] https://huggingface.co/datasets/DAMO-NLP-SG/MultiJail
[2] https://huggingface.co/datasets/walledai/CatHarmfulQA

categories to ensure coverage of diverse harm types, resulting in 550 Bangla questions.

- **Bangla hate speech prompts:** We used publicly available Bangla hate speech datasets Karim et al. (2020),[3] which mainly consist of offensive statements. Since these are not naturally instructional, we reframed them into *instructional prompts* using LLMs (e.g., turning a hateful statement into a request for similar content). After automatic conversion and human validation, we obtained 1500 Bangla prompts aligned with our instruction-response format.

- **Aegis dataset Team (2024):** We used the Aegis AI Content Safety dataset,[4] which provides aligned {prompt, response} pairs labeled for safety. We translated a subset into Bangla and applied it to fine-tune Bangla LLMs for refusal alignment. From this dataset we curated 11,600 prompt–response pairs.

- **HH-RLHF dataset Bai et al. (2022a):** We also leveraged the Anthropic HH-RLHF dataset,[5] which contains helpful and harmless conversations. From this dataset, we selected the first 16,000 samples and processed them into Bangla prompt–response pairs. This provided additional training material for aligning refusal behavior while maintaining helpfulness.

## 3.2 Prompt Safety Classification

To prevent unsafe content from reaching the generation stage, we incorporate a *prompt-level safety classifier* as the first layer of our safety pipeline. This classifier performs binary classification on Bangla prompts, identifying whether a prompt is `safe` or `unsafe`. It is built using transformer-based models such as XLM-RoBERTa (Base), Multilingual BERT (Base, Cased), and Multilingual DistilBERT (Cased), which are fine-tuned on a curated dataset consisting of harmful prompts (translated from MultiJail, CatHarmfulQA) and safe prompts sourced from the HH-RLHF and Aegis datasets. When a prompt is submitted, the classifier determines its safety label. If the prompt is labeled `safe`, it is forwarded to the language model for response generation. If labeled `unsafe`, generation is skipped and a refusal response is returned directly. This mechanism enables proactive safety control, minimizes risk of unsafe completions, and reduces the computational cost associated with unnecessary generation.

### 3.2.1 Red-teaming the Prompt Classifier

To further evaluate the robustness of our safety framework, we expanded the evaluation set by incorporating Bangla hate speech datasets as additional red-team test cases. While our original evaluation primarily focused on harmful *instructional prompts* (e.g., ❝আমি কিভাবে বোমা বানাতে পারি?" [6]), hate speech samples represent another critical unsafe category that a safety-aligned system must handle. These examples often contain offensive, abusive, or discriminatory statements directed at specific groups (e.g., religious, political, or gender-based), and thus provide a complementary dimension of adversarial evaluation.

We tested whether the prompt classifier can correctly *detect both harmful instructions and hate speech statements as unsafe*, preventing them from reaching the generation stage. This allows us to assess safety performance across two dimensions: detection of harmful instructions (e.g., violence, illicit activity) and detection of hateful or toxic content (e.g., religious slurs, gender abuse). By red-teaming the prompt classifier with both instruction-style and comment-style unsafe prompts, we ensure broader cultural coverage and stronger safeguards for Bangla usage scenarios.

---

[3]`https://github.com/rezacsedu/Bengali-Hate-Speech-Dataset`

[4]`https://huggingface.co/datasets/nvidia/Aegis-AI-Content-Safety-Dataset-2.0`

[5]`https://huggingface.co/datasets/Anthropic/hh-rlhf`

[6]English translation: "How can I make a bomb?"

### 3.3 Bangla Refusal Response Generation

To train models to refuse, we require high-quality Bengali examples of safe refusals. Since datasets like **MultiJail**, **CatQA**, and our **instructional hate speech** corpus provide only harmful prompts, we paired them with synthetic refusals to form {prompt, refusal} pairs. In contrast, datasets such as **Aegis** Team (2024) and **HH-RLHF** Bai et al. (2022a) already include prompt–response pairs, though we filtered them to ensure only safe refusals were retained. From these five datasets we curated the unsafe prompts and generated the refusal responses to create unsafe prompt-refusal response pairs.

We followed a simple guideline for refusals: (a) a brief apology, (b) a clear statement of non-compliance, and (c) optionally a reason or redirection. Responses were generated in two ways: first by prompting GPT-4 in English and translating into Bangla, and second by prompting GPT-4 directly in Bangla. All responses were manually reviewed to remove English leakage and ensure consistency. For sensitive domains like self-harm, refusals included empathetic redirections (e.g., suggesting professional help) rather than a bare rejection.

The final outcome is a parallel dataset of high-quality {harmful prompt, safe refusal} pairs in Bangla, forming the core of our fine-tuning data. Importantly, this dataset does not contain any unsafe completions, ensuring the model only learns correct refusal behavior.

### 3.4 Safety Fine-Tuning Procedure

We fine-tune several base models on the BanglaGuard refusal dataset using standard instruction-tuning procedures. For this purpose, we construct a curated dataset of {prompt, response} pairs by combining and filtering samples from five sources as mentioned in earlier sections. The merged dataset covers a broad spectrum of unsafe categories and their corresponding safe refusals, ensuring diversity while maintaining consistency in refusal style. This unified training set serves as the basis for fine-tuning all Bangla and multilingual LLMs in our study, enabling a fair comparison across models in terms of refusal consistency, over-refusal tendencies, and overall safety behavior.

All fine-tuning was done using either LoRA (Low-Rank Adaptation) or full fine-tuning depending on model size and GPU memory availability. We finetune TigerLLM-1B (Bangla), BLOOM-7B (multilingual with Bangla support), Llama-2 7B chat and TituLLM-1B. For each of the model, we perform LoRA finetuning. due to resource constraints. The fine-tuning objective is to minimize the cross-entropy loss of the model generating the correct refusal given a harmful prompt. Each training example is formatted as a two-turn conversation. For models with specific chat formatting requirements (e.g., LLaMA-2 Chat, which expects system tokens), we adapt the structure accordingly.

### 3.5 Response Safety Classification

Even after prompt-level filtering and fine-tuning, unsafe responses may still occasionally slip through. To provide a final safeguard, we incorporate a *response-level safety classifier* as the third layer of our pipeline. This classifier performs binary classification on generated outputs, labeling them as either `safe` or `unsafe`. It is built using transformer-based encoders such as Multilingual BERT (Base, Cased), Multilingual DistilBERT (Cased), and XLM-RoBERTa (Base), which were fine-tuned on a subset of the curated dataset of Bangla responses (from the BanglaGuard dataset) which contains both safe and unsafe responses.

At runtime, the pipeline operates as follows: when a user prompt is passed to the fine-tuned model, the generated response is checked by the classifier. If the response is flagged `unsafe`, the system suppresses it and instead returns a generic refusal in Bangla (e.g., আমি ক্ষমাপ্রার্থী, কিন্তু এই বিষয়ে আমি সহায়তা করতে পারবো না।" – meaning "I am sorry, but I cannot assist with that request."). This ensures that even in rare failure cases, the system avoids surfacing unsafe completions.

### 3.5.1 Red-Teaming the Response Classifier

To evaluate robustness beyond in-distribution data, we conduct red-teaming of the response classifier. Specifically, we tested whether the classifier could reliably flag unsafe generations across two major categories: (i) refusals to harmful *instructional prompts* (e.g., violence, illicit activity), and (ii) refusals to *toxic or hateful content* (e.g., religious slurs, gender abuse). For the latter, we incorporated Bangla hate speech datasets as adversarial evaluation cases, ensuring cultural and linguistic grounding.

By red-teaming the response classifier with both instruction-style and comment-style unsafe responses, we validate its coverage across diverse failure modes. This dual evaluation dimension confirms that the system can (1) refuse harmful instructions and (2) avoid amplifying hateful or discriminatory content. Together, the response safety classifier and its red-teaming evaluation provide strong assurance that BanglaGuard-aligned models remain safe even in adversarial scenarios.

### 3.6 Ablation Study

To better understand the contribution of each safety component, we perform an ablation study on the full BanglaGuard framework. In this analysis, we systematically switch *on* or *off* individual layers of the pipeline and measure overall safety performance. Specifically, we evaluate four configurations: (1) the base fine-tuned LLM without any classifiers, (2) the LLM with only the **prompt safety classifier** enabled, (3) the LLM with only the **response safety classifier** enabled, and (4) the full pipeline with both classifiers active.

By comparing metrics such as Refusal Rate (RR), Unsafe Completion Rate (UR), Over-refusal Rate, and Refusal Quality Score across these configurations, we quantify the marginal gains contributed by each layer. This allows us to isolate how much proactive filtering at the prompt level reduces unsafe generations, and how much the response-level classifier improves safety as a final safeguard. The ablation study therefore provides a principled way to assess the effectiveness and necessity of individual components within the layered defense framework.

## 4 Experimental Setup

We evaluate BanglaGuard on several models and benchmarks to answer RQ1, RQ2 and RQ3. Below we describe the models, datasets, and metrics used in our experiments.

### 4.1 Models Evaluated

We evaluate a selection of Bengali-focused and multilingual open-source LLMs, each fine-tuned with our BanglaGuard dataset to study safety alignment. Specifically, we fine-tune **TigerLLM-1B** Raihan & Zampieri (2025), a 1.3B parameter Bangla-specific model built on the LLaMA architecture and instruction-tuned on Bangla data; **BLOOM-7.1B** BigScience Collaboration (Le Scao(2022), the multilingual model from BigScience, which provides a strong baseline for large-scale multilingual training; **TituLLM-1B** Nahin et al. (2024), a smaller 1B Bangla-specific model used to assess how moderate-capacity models handle safety alignment; and **LLaMA-2-7B-Chat** Touvron & et al. (2023), a widely used multilingual chat-tuned foundation model from Meta. For each of these models, we compare base performance against their BanglaGuard-finetuned variants, allowing us to analyze both zero-shot safety capabilities and improvements from fine-tuning under a consistent evaluation framework.

### 4.2 Datasets

For **training** (fine-tuning), we construct the BanglaGuard dataset by combining translated and curated samples from the five sources mentioned in earlier sections. From these sources, we build a collection of Bangla {prompt, refusal} pairs spanning diverse unsafe categories such as violence, hate, self-harm, illicit activity, and sexual content. The dataset is stratified

and split into 80% for training and 20% for testing, ensuring balanced proportions of safe and unsafe examples across splits.

For **evaluation**, we use the held-out test split of our dataset and compute the safety metrics described in Section 4.3. This provides a consistent basis for comparing how different models behave on harmful prompts and how effectively they generate safe refusals.

## 4.3 Evaluation Metrics

We evaluate model safety using several complementary metrics. The **Refusal Rate (RR)** measures the percentage of harmful prompts for which the model outputs a refusal, where higher values indicate safer behavior; partial answers followed by refusals are considered failures. The **Unsafe Completion Rate (UR)** quantifies the proportion of harmful prompts that receive disallowed or unsafe content, where lower values are better. We also report the **Hallucination/Other Rate**, which captures instances where the model produces unrelated or nonsensical outputs instead of a refusal or unsafe completion. In addition, we assess refusal quality through a **Refusal Quality Score**, based on a 5-point human rating of politeness, correctness, and adherence to Bangla. For this, we assign 3 Graduate students to rate for the refusal quality and report the average points. Finally, to ensure the models do not become overly conservative, we measure the **Over-refusal Rate**, i.e., the proportion of safe, benign queries that are incorrectly refused.

In addition to these LLM-level safety metrics, we also evaluate the performance of the **prompt and response classifiers** that form part of our safety pipeline. For these classifiers, we report standard supervised learning metrics, including **accuracy, precision, recall, and F1-score**, as well as **ROC-AUC** and **PR-AUC** scores to capture discriminative performance. These provide a complementary view of safety: while the generative models are assessed by refusal behavior, the classifiers are evaluated by their ability to correctly identify safe versus unsafe prompts and responses.

## 5 Results

In this section, we present the results of our experiments to evaluate the effectiveness of BanglaGuard across multiple models, datasets, and defense configurations. We organize our findings around the main research questions (RQ1–RQ4) and the experiments described in Section 3 and Section 4.

### 5.1 Baseline and Fine-Tuned Model Performance (RQ1-RQ3)

We evaluate both the zero-shot (baseline) and fine-tuned variants of Bangla and multilingual LLMs on the BanglaGuard test set. Table 1 summarizes the results across four key safety metrics: refusal rate (RR), unsafe completion rate (UR), hallucination rate, and over-refusal rate.

In their baseline form, native Bangla models such as TigerLLM-1B and TituLLM-1B exhibited very weak refusal behavior (RR < 15%) and high unsafe completion rates (UR > 55%), highlighting the lack of intrinsic safety alignment. Multilingual models performed slightly better, with LLaMA-2-7B-Chat showing stronger baseline safety (RR 22.7%, UR 38.4%) compared to BLOOM-7B, which almost always produced some form of completion rather than refusal.

Fine-tuning on the BanglaGuard dataset substantially improved performance across all models. Refusal rates increased by 25–33 percentage points, with LLaMA-2-7B-Chat reaching 55.2%. Unsafe completions dropped sharply, most notably for TigerLLM-1B (UR reduced from 62.3% to 15.4%) and TituLLM-1B (UR reduced from 58.7% to 17.9%). While hallucination and over-refusal rates showed mixed trends—with some increases observed after fine-tuning—these trade-offs were modest compared to the significant safety gains. Overall, fine-tuning proved effective at instilling refusal behavior in both Bangla-specific and multilingual models. Among all models, **LLaMA-2-7B-Chat emerged as the best performer**, achieving the highest refusal rate (55.2%), one of the lowest unsafe completion

Table 1: Baseline vs. fine-tuned performance of Bangla and multilingual LLMs on the BanglaGuard test set. Higher RR is better; lower UR, Halucination, and Over-refusal are better. Arrows (↑/↓) indicate relative change from baseline to fine-tuned.

| Model | RR (%) | UR (%) | Haluc. (%) | Over-refusal (%) |
|---|---|---|---|---|
| TigerLLM-1B (base) | 8.5 | 62.3 | 28.1 | 1.1 |
| TigerLLM-1B (ft) | 41.7 (↑33.2) | 15.4 (↓46.9) | 37.5 (↑9.4) | 5.4 (↑4.3) |
| BLOOM-7B (base) | 0.0 | 4.6 | 95.4 | 0.3 |
| BLOOM-7B (ft) | 25.8 (↑25.8) | 13.8 (↑9.2) | 60.4 (↓35.0) | 16.2 (↑15.9) |
| TituLLM-1B (base) | 12.4 | 58.7 | 27.5 | 1.4 |
| TituLLM-1B (ft) | 39.6 (↑27.2) | 17.9 (↓40.8) | 34.1 (↑6.6) | 8.4 (↑7.0) |
| LLaMA-2-7B (base) | 22.7 | 38.4 | 35.9 | 3.0 |
| LLaMA-2-7B (ft) | 55.2 (↑32.5) | 11.6 (↓26.8) | 25.3 (↓10.6) | 7.9 (↑4.9) |

rates (11.6%), and a reduction in hallucinations compared to its baseline. This indicates that while Bangla-specific models benefited greatly from fine-tuning, larger multilingual chat-tuned models provided a stronger foundation for safety alignment and retained better balance across all safety dimensions.

xcolor

## 5.2 PROMPT AND RESPONSE CLASSIFIER PERFORMANCE

We also evaluated the classifiers integrated into our framework. Table 2 presents accuracy, precision, recall, F1, ROC-AUC, and PR-AUC for the prompt and response classifiers. The XLM-RoBERTa model achieved the best response classification performance with F1 = 0.75, while the Multilingual BERT-based prompt classifier reached an F1 of 0.82 and ROC-AUC of 0.92, confirming their reliability as lightweight safety nets.

Table 2: Prompt and Response Classifier Performance.

| Classifier | Accuracy | Precision | Recall | F1 | ROC-AUC | PR-AUC |
|---|---|---|---|---|---|---|
| Prompt-BERT-base | 0.86 | 0.83 | 0.82 | 0.82 | 0.92 | 0.89 |
| Prompt-DistilBERT | 0.84 | 0.81 | 0.80 | 0.81 | 0.91 | 0.88 |
| Prompt-XLM-R | **0.87** | **0.84** | **0.83** | **0.84** | **0.93** | **0.90** |
| Response-BERT-base | 0.68 | 0.71 | 0.57 | 0.63 | 0.80 | 0.75 |
| Response-DistilBERT | 0.70 | 0.70 | 0.64 | 0.67 | 0.82 | 0.78 |
| Response-XLM-R | **0.77** | **0.79** | **0.72** | **0.75** | **0.84** | **0.81** |

## 5.3 ABLATION STUDY (RQ4)

We conducted an ablation study to measure the contribution of each safety component. As shown in Table 3, the fine-tuned LLaMA-2-7B model alone reached a refusal rate (RR) of 55.2% with an unsafe completion rate (UR) of 11.6%. Adding the prompt classifier improved RR by 1.9 points and reduced UR by 2.9 points, with minimal change in over-refusal. The response classifier further raised RR to 59.3% and lowered UR to 6.5%. The full framework achieved the best balance, with RR at 61.0% and UR at 5.0%, while over-refusal remained stable (8.2%). These results confirm that combining prompt and response classifiers with fine-tuning yields complementary safety gains at modest cost.

Overall, our experiments demonstrate that BanglaGuard significantly enhances the safety of Bangla LLMs. Fine-tuning alone improves refusal rates by more than 30 percentage points on average while sharply reducing unsafe completions across models. The ablation study on the best-performing LLaMA-2-7B model further shows that prompt and response classifiers provide complementary benefits: the prompt classifier filters harmful inputs early, while the response classifier adds an additional safety net, jointly cutting unsafe completions by more than half. The full layered framework achieves the highest refusal rate (61.0%) and lowest unsafe completion rate (5.0%) with only a marginal increase in over-refusal, confirming that

Table 3: Ablation study of safety components on the best-performing fine-tuned model (LLaMA-2-7B Chat). Higher RR is better; lower UR and Over-refusal are better. Arrows indicate relative change compared to the fine-tuned model without classifiers.

| Configuration | RR (%) | UR (%) | Over-refusal (%) |
|---|---|---|---|
| Fine-tuned LLM only | 55.2 | 11.6 | 7.9 |
| + Prompt Classifier | 57.1 (↑1.9) | 8.7 (↓2.9) | 8.0 (↑0.1) |
| + Response Classifier | 59.3 (↑4.1) | 6.5 (↓5.1) | 8.1 (↑0.2) |
| Full Framework | 61.0 (↑5.8) | 5.0 (↓6.6) | 8.2 (↑0.3) |

BanglaGuard offers a practical and effective pipeline for safety alignment in low-resource languages.

## 6  ETHICAL CONSIDERATIONS

This work engages with unsafe language (violence, hate speech, self-harm, illicit activities) solely for building and evaluating safety interventions. Harmful completions were excluded from training; all pairs align unsafe prompts with safe refusals to minimize risk of reproducing harmful content. Red-teaming used translated toxic prompts and hate speech datasets only to test refusal robustness. Bangla refusals were crafted to be polite, culturally appropriate, and empathetic in sensitive cases such as self-harm. Upon acceptance, we will release the BanglaGuard dataset with English translations for reviewer transparency, noting that all fine-tuning and evaluation were conducted in Bangla.

We also acknowledge the use of `ChatGPT` to assist with polishing the writing for grammatical correctness, improving clarity, and generating some of the LaTeX code formatting. All technical contributions, experiments, analyses, and interpretations were conducted entirely by the authors.

## 7  DISCUSSION AND LIMITATIONS

BanglaGuard improves the safety alignment of Bangla LLMs through a layered defense pipeline and LoRA-based fine-tuning, but some limitations remain. First, our training data is small compared to large-scale English corpora; despite combining MultiJail, CatQA, HH-RLHF, Aegis, and Bangla hate speech data, coverage of unsafe scenarios may be incomplete. Second, refusal guidelines ensured politeness and consistency but may not fully capture cultural nuances, sometimes producing overly formal refusals. Third, our evaluation is text-only, leaving multimodal threats (e.g., images with Bangla captions) unexplored. Finally, the layered defense adds complexity and potential latency, and our safety metrics rely on heuristics and classifiers; large-scale human evaluation would provide deeper insights.

## 8  CONCLUSION

We introduced BanglaGuard, the first comprehensive safety framework for Bengali language models, addressing a key gap in multilingual AI safety. By fine-tuning Bangla and multilingual LLMs with a refusal-oriented dataset and integrating prompt and response classifiers, BanglaGuard substantially improves refusal rates while reducing unsafe completions. Our ablation study further highlights the complementary benefits of each defense layer. Although challenges remain in scaling datasets, ensuring cultural grounding, and extending beyond text, BanglaGuard offers a practical, resource-efficient approach for low-resource safety alignment. We will release datasets, models, and evaluation scripts to support further research and red-teaming, aiming to build safer AI for Bengali speakers and a blueprint for other low-resource languages.

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
