# OpenReview forum: "BanglaGuard: Benchmarking and Defending Large Language Models for Safety in Low-Resource Languages"
_ICLR.cc/2026/Conference — Submitted to ICLR 2026_

### Official Review · Reviewer_Saai · 2025-10-31

**Soundness:** 2
**Presentation:** 1
**Contribution:** 1
**Rating:** 2
**Confidence:** 5

**Summary:**

The paper introduces BanglaGuard, a safety framework for Bengali large language models (LLMs), comprising a dataset of 29,950 safe/unsafe Bangla prompts and a three-tier defense pipeline (prompt classification, LoRA-based fine-tuning, and response classification). Experiments show that fine-tuning improves refusal rates by 25–33 points and reduces unsafe completions, with LLaMA-2-7B-Chat achieving a 61.0% refusal rate and 5.0% unsafe completions. The work positions itself as the first comprehensive safety solution for Bangla.

**Strengths:**

1. The curated Bangla safety dataset (29,950 prompts) fills a critical gap for low-resource languages, enabling future research.

2. Refusal responses are tailored to Bangla linguistic norms, and hate speech datasets are incorporated for culturally relevant red-teaming.

**Weaknesses:**

1. The core methodology (e.g., prompt/response classifiers, LoRA fine-tuning) is a direct application of existing safety techniques to Bangla, without algorithmic or theoretical innovation. It is more like a technical report than a scientific research paper, as if it is an engineering implementation of existing safety technologies in Bengali, lacking hypothesis-driven research or comparative analysis against state-of-the-art multilingual safety methods

2. The article is difficult to be considered as effectively answering the RQ1-RQ4 in the Introduction. These issues are only mentioned in a section of the experiment and are not analyzed individually.

3. The article does not conduct targeted analysis of the challenges and issues faced by small languages with resource scarcity, but instead uses the pattern of common languages to collect new data for small languages, which does not provide new inspiration for future work.

**Questions:**

Same as Weakness

---

### Official Review · Reviewer_Fq9e · 2025-10-31

**Soundness:** 1
**Presentation:** 1
**Contribution:** 1
**Rating:** 2
**Confidence:** 5

**Summary:**

The paper introduces BanglaGuard, the safety benchmark and defense pipeline for Bengali LLMs. It builds a dataset of harmful/safe prompts and refusal pairs, and applies a three-layer framework to improve safety.

**Strengths:**

- The proposed three-tier system—prompt classifier, fine-tuned model, and response classifier—offers a  empirically validated method to enhance model safety.

- Fine-tuning led to safety improvements with minimal loss in helpfulness, showing methodological rigor and balanced trade-offs.

**Weaknesses:**

- What authors mean by Bengali Large Language Models (written in abstract)?

- It isn’t clear why the authors selected the specific datasets they mention, given that existing datasets already cover Bengali with harmful content labels and include cultural context. See the datasets https://huggingface.co/datasets/SoftMINER-Group/Soteria , https://github.com/NeuralSentinel/CulturalKaleidoscope , https://huggingface.co/datasets/zhiyuan-ning/linguasafe

- The related work section appears to omit significant prior research on Bengali AI safety and culturally sensitive datasets. https://aclanthology.org/2025.naacl-long.388/ , https://arxiv.org/abs/2502.11244 , https://huggingface.co/datasets/SoftMINER-Group/Soteria , https://arxiv.org/pdf/2508.12733v1 . The authors should acknowledge these works and ideally perform experiments using those datasets as well.

- The dataset relies heavily on translated English sources, lacking genuine Bangla-specific harmful prompts, which reduces cultural and linguistic representativeness.

- Testing on a held-out split of its own dataset risks overfitting, meaning results may not generalize well to unseen or adversarial prompts.

- The framework ignores Bangla-English code-switching and Romanized Bangla, making it vulnerable to real-world mixed-language inputs.

-  The analysis of helpfulness post-safety tuning is shallow; over-refusal tendencies are underexplored and could reduce usability.

- The dual-classifier setup increases potential failure points and latency, relying on heuristics that may not be robust under adversarial or obfuscated inputs.

- Authors Claims that BanglaGuard can serve as a blueprint for other low-resource languages are unsupported, as no cross-lingual or multimodal validation is presented.


**Minor comments**

- The overall writing quality could be improved for coherence and professional presentation.

**Questions:**

See weaknesses

---

### Official Review · Reviewer_zgb6 · 2025-10-31

**Soundness:** 2
**Presentation:** 1
**Contribution:** 2
**Rating:** 2
**Confidence:** 4

**Summary:**

This paper presents BanglaGuard, a safety framework for Bengali large language models. The authors first introduce a new dataset of nearly 30k Bangla prompt–response pairs, created by translating and combining several existing English datasets and manually reviewing them for cultural appropriateness. On top of this, they build a safeguard pipeline consisting of three modules, namely a prompt classifier, LoRA-based fine-tuning, and a response classifier. Experiments are conducted on several LLMs. The authors argue that BanglaGuard can serve as a scalable template for safety alignment in low-resource languages.

**Strengths:**

- An extensive dataset compilation combining multiple multilingual sources and manual review.
- Ablation study shows complementary benefits of combining classifiers and fine-tuning.

**Weaknesses:**

- Line 454–455: The authors state, “Upon acceptance, we will release the BanglaGuard dataset with English translations for reviewer transparency.” However, to improve transparency and reproducibility during the review process, it would be preferable to provide at least partial or anonymized access to the dataset prior to acceptance.
- Limited novelty: three-stage pipeline (prompt classifier, fine-tuned LLM, response classifier) is conceptually straightforward.
- The dataset composition and terminology remain somewhat unclear. While Section 3.1 lists several data sources (MultiJail, CatHarmfulQA, HH-RLHF, Aegis, and Bangla hate-speech corpora), parts of the composition description are inconsistent. For example, the authors state that the MultiJail dataset contains 270 harmful prompts (15 scenarios × 18 languages), yet they report curating 300 Bangla prompts from it. It is not explained how more prompts were obtained than the original dataset provides. Furthermore, the included datasets differ in structure: some offer only prompts (e.g., MultiJail, CatHarmfulQA, hate-speech corpora), whereas others provide prompt–response pairs (Aegis, HH-RLHF). Consequently, the merged dataset contains both prompts and refusals, making the name “Bangla Unsafe Prompt Dataset” somewhat misleading. A clearer breakdown of individual dataset sizes, preprocessing steps (translation pipeline, filtering, and manual validation), and the procedure for merging them into the final 29,950 samples would substantially improve transparency and reproducibility.
- Lack of transparency in evaluation methodology:
    - unclear red-teaming procedure (selection, generation, criteria)
    - missing equations and detailed definitions for all reported metrics (refusal, unsafe completion, hallucination, over-refusal, quality). Therefore, it is unclear how hallucination and refusal quality metrics are computed.
    - The Refusal Quality metric is mentioned in the experimental setup, yet corresponding results are not reported in the results section (Table 1).
- Within the experiments, no cross-validation or statistical analysis of variance across random seeds or raters is provided

**Questions:**

1. What is meant by “instruction–response format” (line 169)?
2. Could the authors provide formulas or pseudocode for all reported metrics, especially the “Hallucination/Other Rate” and “Refusal Quality Score”?
3. How was the red-teaming conducted? Which datasets, prompt sampling, and attack formats were used?
4. Given that MultiJail includes 270 prompts (15 scenarios × 18 languages), how were 300 Bangla prompts curated from it?

---

### Official Review · Reviewer_oNpk · 2025-11-02

**Soundness:** 3
**Presentation:** 3
**Contribution:** 2
**Rating:** 2
**Confidence:** 4

**Summary:**

This paper introduces BanglaGuard, a safety framework for Bengali LLMs. BanglaGuard aims to address comparatively high rates of unsafe content generation in existing LLMs.
The authors introduce a new benchmark dataset of 29,950 Bangla prompts paired with refusal responses. Subsequently they build a three-tier defense pipeline combining a prompt classifier, a safety-tuned LLM, and a response classifier to filter unsafe content. Experiments show that the model meaningfully reduces generation of unsafe content.

**Strengths:**

- novel and practical dataset. the BanglaGuard dataset makes for a valuable contribution
- paper considers all stages of safety mitigation (input/ouput filtering and model alignment) Approach makes improvements to each stage
- authors empirically demonstrate reduction in unsafe content generation

**Weaknesses:**

# Major

- The paper is missing a discussion and definition of the safety taxonomy it operates on. Currently the authors simply mix 5 datasets without discussing what they deem to be safe or unsafe content in the first place. This makes for a rather shallow safety notion, since a reasonable taxonomy should directly inform the data collection and any annotation efforts.
- Along these lines the paper claims to "compose culturally relevant prompts", but I fail to see this contribution in the respective section. However, a more culturally sensitive approach would strongly benefit the paper. By largely translating existing English datasets the authors rely on western-centric notions of safety which are likely to inaccurately capture certain Bengali safety norms
- Treating safety as a binary classification problem by only discerning between safe/unsafe content does not accurately reflect the more gradual scale that is perception of safety. Especially, due to the lack of a safety taxonomy this makes decision boundaries rather arbitrary. Further, for socially relevant context a distinction between outright unsafe or culturally sensitive content would be better (see for example parallel ICLR submission: https://openreview.net/forum?id=ScVl9QlLpD)
- The paper is generally missing a comprehensive evaluation on potential trade-offs in overall helpfulness/usefulness vs safety. A trivially safe system simply refuses to answer any question. The only notion on this aspect is the reported over-refusal score, but the paper needs more comprehensive benchmarks to accurately assess the effect of this guardrail system
- small training corpus. As acknowledged by the authors themselves the BanglaGuard dataset makes for a sizable benchmark but is rather small as a training dataset
- this also leads leads to reduced performance gains with the best combined system only achieving a 61% RR. At the same time we see a potential trade-off in training of this data with 3 out of 4 models showing a decrease in UR/Halucination respectively. All models also slightly increase over-refusal


- In general the paper lacks a lot of detail in both the methodological description and result analysis. There does not seem to be any Appendix that would cover this laps. For example, further details on the exact makeup of the dataset, its categories, distributions of prompts/answers of safe/unsafe content and categories are missing. No detailed information is provided about the translation and human annotation process. From the description it is not even clear that all datasets are processed the same (e.g. no human validation mentioned for Aegis). Model evaluations lack the same details and we are given no insights/anaylsis on failure cases, whether there are systematic shortcoming in current approaches. Also no existing safeguard are evaluated (simple baseline could be multilingual LLMs that are instructed with the safety taxonomy)

- Paper does not provide a data sheet for the introduced dataset (https://arxiv.org/abs/1803.09010)


## Minor comments
- Figure 1 is hard to read and general presentation here could be improved
- the authors introduce a Refusal Quality Score which is never reported in any of their results
- "xcolor" in line 396

**Questions:**

- the authors state that "All fine-tuning was done using either LoRA (Low-Rank Adaptation) or full fine-tuning depending on model size and GPU memory availability." in the same paragraph they also write "For each of the model, we perform LoRA finetuning. due to resource constraints." Does that mean you only perform LoRA finetuning? Why the claim about full fine-tuning then?
- Can the authors provide further details on the pipeline for dataset curation?
- Can the authors provide more insights on current failure cases of the different safeguard stages? Are there any systematic shortcomings or performance differences in certain categories?


Please see also my comments on weakness

**Details Of Ethics Concerns:**

The paper uses human annotators to label safety related content. Unfortunately, the authors provide no information on the compensation of annotators, measures to ensure ethically responsible exposure to unsafe material, whether ERB approval was obtained, etc.
Further no details on the actual study setup are provided other then relying on 3 graduate students.

---

### Official Review · Reviewer_6RPx · 2025-11-03

**Soundness:** 3
**Presentation:** 3
**Contribution:** 3
**Rating:** 6
**Confidence:** 3

**Summary:**

This work deals with making LLMs safer by developing a mechanism to prevent them from answering dangerous queries. Specifically, the authors address this task for Bengali, which is a language underrepresented in both data and LLM research. They address this problem through a 3-step framework, BanglaGuard, which reduces unsafe responses in Bengali LLMs by (1) identifying unsafe prompts using a text classifier, (2) fine-tuning the LLM to reject unsafe prompts, and (3) identifying unsafe responses using another text classifier. To train and evaluate this framework, the authors create a dataset of prompts paired with safe/unsafe labels. This dataset was created by combining 5 sources, translating English prompts into Bengali and processing the Bangla Hate Speech corpus. For evaluation, they consider 4 open-source LLMs spanning multilingual and Bengali language-specific models. They compare the models on their refusal rate (RR), unsafe completion rate (UR), and over-refusal rate (OR). These are comparable to true positive, False negative, and false positive rates. Additionally, they also evaluate these models on the refusal quality score, based on 3 human evaluators’ rating of LLM responses refusing to answer an unsafe query. They find that while off-the-shelf open-source LLMs perform poorly on these metrics, fine-tuning them and combining them with the prompt and response classifiers drastically improves their ability to avoid unsafe responses.

**Strengths:**

- This work addresses an important problem: safety in Bengali language models
- The authors contribute a new dataset of Bengali prompts with safe/unsafe labels to evaluate and finetune language
- Their empirical results indicate that their proposed 3-stage framework reduces the possibility of unsafe responses.
- The manuscript is clear and well-written

**Weaknesses:**

- The reported results appear to be over single training runs. Since neural network training is stochastic, it is unclear how much of the difference in results is attributable to that stochasticity. Reporting the mean and standard deviation over multiple runs would make this clear.

- The proposed framework relies on three loosely coupled components, which would increase latency. This might be addressed through tighter integration of steering mechanisms, e.g., in Härle et al. (2025), Loula et. al. (2025), and Ahmed et al. (2025).

- Minor: The Bengali text examples on pages 4 and 5 seem to have formatting errors.


References:

Ahmed et al. "Controllable Generation via Locally Constrained Resampling." ICLR (2025)

Härle et al. “Measuring and Guiding Monosemanticity.” NeurIPS (2025)

Loula et al. "Syntactic and Semantic Control of Large Language Models via Sequential Monte Carlo." ICLR (2025).

**Questions:**

How do proprietary models like GPT-4o perform on this data set?

---

### Meta-Review · Area_Chair_cXaQ · 2025-12-12

**Summary:**

There are several major concerns raised by the reviewers: 1. The novelty of the proposed framework is limited, as it primarily combines existing techniques without significant innovation. 2. Certain claims of the proposed framework lack empirical support. 3. Some important baselines or comparative experiments are missing. 4. The experimental results are based on a signal run, whichundermines their reliability.

**Reviewer Concerns:**

The authors did not submit a rebuttal. Therefore, all concerns are still outstanding.

**Reviewer Scores:**

The authors did not submit a rebuttal. I do not think the reviewers would change their scores.

---

### Decision · Program_Chairs · 2026-01-26

Reject